# Anti-Biofilm Effect of Bacteriophages and Antibiotics against Uropathogenic *Escherichia coli*

**DOI:** 10.3390/antibiotics11121706

**Published:** 2022-11-26

**Authors:** Laima Mukane, Karlis Racenis, Dace Rezevska, Aivars Petersons, Juta Kroica

**Affiliations:** 1Department of Biology and Microbiology, Riga Stradins University, LV-1007 Riga, Latvia; 2Center of Nephrology, Pauls Stradins Clinical University Hospital, LV-1002 Riga, Latvia; 3Department of Internal Diseases, Riga Stradins University, LV-1007 Riga, Latvia; 4Joint Laboratory, Pauls Stradins Clinical University Hospital, LV-1002 Riga, Latvia

**Keywords:** *E. coli*, biofilm, urinary tract infection, phage, phage adaptation, MBEC

## Abstract

*Escherichia coli* is a common cause of biofilm-associated urinary tract infections. Bacteria inside the biofilm are more resistant to antibiotics. Six *E. coli* strains isolated from patients with urinary tract infections were screened for biofilm-forming capability and antimicrobial susceptibility. Two of the most significant biofilm-producing strains were selected for minimal inhibitory concentration and minimal biofilm eradication concentration in vitro testing using amoxicillin–clavulanic acid, ciprofloxacin, and three commercial bacteriophage cocktails (Pyobacteriophag, Ses, and Intesti). In case of a low phage effect, an adaptation procedure was performed. Although the biofilms formed by strain 021UR were resistant to amoxicillin–clavulanic acid and ciprofloxacin, the three phage cocktails were able to reduce biofilm formation. In contrast, phages did not affect the 01206UR strain against planktonic and biofilm-forming cells. After Pyobacteriophag adaptation, the effect improved, and, regardless of the concentration, the adapted phage cocktail could destroy both planktonic cells and the biofilm of strain 01206UR. Bacteriophages capable of killing bacteria in biofilms can be used as an alternative to antibiotics. However, each case should be considered individually due to the lack of clinical trials for phage therapy. Antimicrobial and phage susceptibility should be determined in biofilm models before treatment to achieve the desired anti-biofilm effect.

## 1. Introduction

Antibiotic resistance is spreading rapidly and is a major problem for the healthcare system. The World Health Organization (WHO) Global Antimicrobial Resistance and Use Surveillance System (GLASS) report [1] confirms that antibacterial resistance is increasing, specifically in low- and middle-income countries, causing significant mortality and morbidity. New and wide-spectrum antibiotics are rarely invented, and their costs are usually high [2,3]. Since the first analysis in 2017, only two approved agents represent new chemical classes. Most recently, authorized agents are derivatives of known classes. There are 217 antibacterial agents/programs that are in the preclinical stage. The annual analysis of the preclinical pipeline shows that from one year to the other, one-third of the development programs are discontinued [4].

Treatment with antibiotics can also cause unwanted side effects and even toxic effects for the patient, which can be very different and depend on the specific active substance and the group of drugs represented, including hypersensitivity reactions, gastrointestinal disturbances, and even nephro- or neurotoxicity [5]. Bacteriophages, which, unlike antibiotics, do not cause unwanted antibiotic-associated side effects, can be used as an alternative therapy. For example, in cases where the patient has kidney failure and the only effective antibiotics are nephrotoxic, it is necessary to look for alternatives, so that they do not cause irreversible damage to the patient [6]. 

Urinary tract infections (UTIs) are common infections that require antibacterial therapy, which can cause changes in the microbiome and promote the development of multidrug-resistant bacterial strains [7]. *E. coli* is one of the most common causes of UTIs [7,8,9,10]. It is a Gram-negative rod-shaped bacteria with flagella, which help with movement and provide the adhesion function [11]. Resistance development is facilitated by the ability of bacteria to form biofilms, which make the microorganism more virulent and help to resist the host’s immune response by enclosing them in an extracellular biochemical matrix. Biofilms are associated with up to 60% of human infections [12]. They can form in human tissues on altered or damaged surface structures. Biofilms are often related to the ability of bacteria to adhere to surfaces, and, in this case, the morphology of the bacteria plays an important role. *E. coli* has great potential to form a biofilm in the case of a UTI [13]. Studies show that the ability to create a biofilm depends on various factors. One of them is that biofilm formation is genetically encoded and is one of the defense mechanisms that ensure resistance [14,15]. Many studies indicate that the mechanical and physical–chemical properties of the biofilm matrix reduce or delay the penetration of many compounds, including antibacterial agents. As a result, UTIs are becoming increasingly difficult to treat. The WHO has listed resistant *Enterobacterales* as one of the global priorities for the research and development of new antibacterial agents [1].

The use of antibiotics is commonly based on their effect on planktonic bacterial cells, without paying enough attention to the risk of biofilm formation. Some antibiotics, such as fluoroquinolones, rifampin, and ampicillin, penetrate well through the matrix, although they do not completely eradicate the biofilm. However, delayed antibiotic penetration may have critical phenotypic consequences. Due to the slow diffusion of antibiotics through the biofilm, bacteria could be exposed to long-term subinhibitory concentrations of antibiotics. Subsequently, bacterial cells could adapt to the presence of antibiotics through metabolic or transcriptional adaptation [16]. There is evidence that sub-inhibitory concentrations of antibiotics can even promote the formation of bacterial biofilms [17,18].

Phages have been explored as antibiofilm agents, as they are capable of lysing host bacteria in biofilms. Although the efficacy of these phages is limited [19], phages can usually infect and destroy only a subset of strains belonging to a single bacterial species. Therefore, several methods can be used to expand the spectrum of phage efficacy and reduce the risk of resistance. Using a mixture of several lytic phages as a phage cocktail can overcome the limitations of monotherapy and improve treatment results [20]. Recent studies have shown that a phage cocktail is more effective at preventing and destroying bacteria biofilms than individual phages [21]. Another option to improve the effect against resistant strains is to perform an adaptation procedure in which phages are trained to infect the host [22]. 

In this study, we tested the biofilm-formation capacity of *E. coli* strains isolated from urine samples from patients with a UTI. The antimicrobial and anti-biofilm effect of amoxicillin–clavulanic acid, ciprofloxacin, and three commercial bacteriophage cocktails was detected in the two strongest biofilm-forming strains. In the event of low phage efficiency, a bacteriophage-adaptation procedure was performed, and the improvement of the lytic effect was evaluated.

## 2. Results

### 2.1. Bacterial Isolates

Six *E. coli* strains included in the study were isolated from five female and one male patients with different types of UTI and comorbidities, as shown in Table 1.

### 2.2. Biofilm Screening

The *E. coli* strain 01108UR did not produce any biofilm because the optical density was similar to that of the negative control. Five strains showed different biofilm-formation capabilities (see Figure 1). The two strains (021UR and 01206UR) with the highest optical density values were selected for further MIC and MBEC testing.

### 2.3. Antimicrobial and Phage Susceptibility

The original titer of the bacteriophage cocktail Ses was 4 × 10^5^ PFU/mL, Intesti was 7 × 10^5^ PFU/mL, and Pyobacteriophag was 3 × 10^6^ PFU/mL. After propagation of the phage cocktail, the titer increased to 4 × 10^6^, 1 × 10^8^, and 4 × 10^6^ PFU/mL for Ses, Intesti, and Pyobacteriophag, respectively. The data on the susceptibility of bacterial strains to phages are summarized in Table 2. We observed that the strain 021UR was more sensitive to all phage cocktails, having semi-confluent lysis in the case of Ses and Pyobacteriophag and confluent lysis when the Intesti phage cocktail was applied. The lowest sensitivity of the phage cocktails was observed for 01108UR and 01206UR. The strain 0108UR showed resistance to Intesti and Pyobacteriophag and partial lysis to Ses. The strain 01206UR showed resistance to the Ses and Intesti phage cocktails and partial lysis to Pyobacteriophag. Phage adaptation was performed to improve the efficiency of the Pyobacteriophag cocktail with 01206UR, as this strain was chosen for further antibiofilm testing. After adaptation, the Pyobacteriophag titer increased to 7 × 10^7^ PFU/mL. The phage titer was equalized to 10^6^ PFU/mL to perform the spot assay for phage-susceptibility testing. Improvement of the Pyobacteriophag effect in strain 01206 from individual plaques (+) to confluent lysis (+++) was detected after adaptation (Table 2 and Figure 2).

Bacterial strains showed different antimicrobial susceptibilities (see Table 2). Of the six bacterial strains, 01081UR and 01108UR were susceptible to all antimicrobials tested; 01032UR showed the broadest antimicrobial resistance, being resistant to six antimicrobials; 021UR and 01206UR were resistant to three antimicrobials; and 01168UR was resistant to two antimicrobials.

### 2.4. Colony-Forming Unit Spot Test

The results of the tests are reflected in Table 3 and visualization of the test is represented in Figure 3. According to the results, even after 24 h, the biofilm continued to form, and, after 48 h, the number of bacteria increased.

### 2.5. Minimal Inhibitory Concentration (MIC) and Minimal Biofilm Eradication Concentration (MBEC) Detection

According to EUCASTv9.0, the MIC values of amoxicillin–clavulanic acid and ciprofloxacin are 8 mg/L and 0.25 mg/L, respectively. The strain 021UR was resistant to amoxicillin–clavulanic acid, and the MIC value was reached only at a concentration of 256 mg/L, but MBEC was not reached even at the highest concentration (1024 mg/L). The MIC value of ciprofloxacin was achieved at 0.25 mg/L but showed resistance in the biofilm; MBEC was 64 mg/L. Although strain 01206UR was susceptible to amoxicillin–clavulanic acid, and the MIC value was reached at 8 mg/mL, this antibiotic failed to eradicate the bacterial biofilm at a concentration of 1024 mg/L. The strain 01206UR was also resistant to ciprofloxacin; the MIC value was 128 mg/L, but MBEC was not reached at 1024 mg/L. The mean MIC and MBEC values in the test of amoxicillin–clavulanic acid and ciprofloxacin are represented in Figure 4. Both antibiotics show no inhibiting effect on biofilms. 

The mean values of the MIC and MBEC tests of the phages are represented in Figure 5 and Figure 6. We conclude that a strain was sensitive to a phage if the OD values were <0.1 measured at 650 nm or if the result was close to the negative control. Using phages, the growth of bacterial strain 021UR was reduced in all cases, including the planktonic cells (MIC) and biofilms (MBEC), compared to the positive control, but the effect is not adequate enough in all cases to declare that the strain is completely lysed by the phage cocktail, as the growth of bacteria is not observed. The other strain, 01206UR, showed resistance to all three phage cocktails (Pyobacteriophag, Ses, and Intesti). Resistance to the Ses and Intesti phage stocks was already determined in the phage-susceptibility test. However, for Pyobacteriophag, we observed a low lytic effect, so the adaptation of Pyobacteriophag was performed. After that, efficiency improved: bacterial growth was markedly reduced in both planktonic (MIC) and biofilm-forming (MBEC) cells. However, the OD values were slightly above 0.1 for the MIC.

## 3. Discussion

Our data indicate that *E. coli* isolated from patients with a UTI can often form biofilms. The results show that although amoxicillin–clavulanic acid and ciprofloxacin can successfully deal with planktonic cells, they cannot destroy biofilms. In contrast to antibiotics, our research demonstrates that if bacteriophages affect planktonic cells, they can also destroy biofilms. In situations where the bacteriophages initially showed a weak effect against the strain, phages could be adapted, thus achieving a lytic effect in both planktonic and biofilm-forming cells.

*E. coli* is often associated with biofilm formation [23,24,25,26], although our research shows that the ability of bacteria to form a biofilm is highly variable depending on each individual strain. In the human body, bacteria can attach to tissues and to artificial surfaces, such as catheters and implants, to form a biofilm [11]. It is self-evident that biofilms, as more resistant forms of bacteria, will often be found in samples of patients with contributing factors, such as comorbidities, weakened immunity, and artificial devices, for example, urinary catheters. Referring to this, in our study, the strongest biofilm-producing strain, 01206UR, was collected from a patient with a transplanted kidney, chronic kidney disease, secondary hypertension, and a recurrent UTI. However, interestingly, the results reveal that the second-strongest biofilm-forming strain, 021UR, was isolated from a patient with pyelonephritis and without any other comorbidities. We did not investigate the genetic determinants of the formation of bacterial biofilms, but this would be essential to fully understand bacterial biofilms. Highly resistant bacterial strains are not necessarily the strongest biofilm formers [27,28]. For instance, the ESBL-positive 01032UR strain described in our study was resistant to amoxicillin, amoxicillin–clavulanic acid, cefotaxime, and ceftazidime. Still, compared to the other strains, it was a weak biofilm producer. The strongest biofilm producers were 021UR and 01206UR; these strains were resistant to three antibiotics. The ability of bacteria to form biofilm should be considered an essential aspect during infection and treatment. Before starting therapy, bacteria isolated from patient material should be tested for antimicrobial susceptibility and biofilm-formation ability.

Amoxicillin–clavulanic acid and ciprofloxacin are common antibiotics used for the treatment of UTIs. Referring to the antibiotic sensitivity test, the bacterial strain 021UR was resistant to amoxicillin–clavulanic acid, which was also confirmed by the MIC test in a 96-well plate; therefore, it is not unexpected that amoxicillin–clavulanic acid was unable to destroy the biofilm in this strain. The results are different for the 01206UR strain, although the bacterial strain is sensitive to amoxicillin–clavulanic acid, and the MIC value is reached at a low concentration (8 mg/L); in the case of biofilm, even at the maximum concentration (1024 mg/L), the antibiotic was unable to destroy the biofilm. Ciprofloxacin showed similar results. The strain 01206UR, determined to be resistant to ciprofloxacin, showed a high MIC value at 128 mg/L and did not destroy the biofilm even at 1024 mg/L. Ciprofloxacin showed remarkable results for the 021UR strain, i.e., although planktonic cells are destroyed, and a MIC value is reached at a very low concentration (0.25 m/L), ciprofloxacin was able to destroy the biofilm only at a much higher concentration (64 mg/L). Planktonic cells can be eradicated in the human body through the combined action of antimicrobials and host immune responses. Although antibiotics can successfully destroy planktonic bacterial cells, they are often unable to destroy biofilms. Bacteria in the biofilm often survive treatment and can cause reactivation of the infection [16]. Additionally, antibacterial treatment can suppress the normal microflora and promote the growth of other bacteria. 

Occasionally, to achieve MIC and MBEC values in biofilm models, it is necessary to use high concentrations of antibiotics [18]. Such high doses cannot be used for patients because they can be harmful or even fatal to the patient; therefore, such antibiotics will not provide an effect in therapeutic doses and are not recommended for treatment. Our results show that phages can achieve a better effect on biofilms (MBEC values) than on planktonic cells (MIC values). This can be observed most clearly when analyzing the effect of the SES phage against the 021UR strain (see Figure 6) and could be explained by the fact that phages begin active replication when bacteria are released from the biofilm and become metabolically active [29]. The three phage cocktails could disrupt bacterial biofilms for the 021UR strain. However, this strain was resistant to amoxicillin–clavulanic acid, and, for ciprofloxacin, the MBEC value was reached only at a concentration of 64 mg/L. 

Interestingly, the results show that although both strains, 021UR and 01206UR, are strong biofilm formers and express resistance to several antibiotics (see Table 2), they are still susceptible to phages. As a limitation, it should be mentioned that our study did not use individual phages but used commercial phage cocktails. Using commercial phage cocktails, it is difficult to multiply them because we do not have access to the original host strains on which these phages were grown and propagated. Unlike antibiotics, phages are living viruses, so their activity can be affected by various factors, such as the amount of nutrients, temperature, and pH. Therefore, it is critical to provide appropriate growth conditions during research. The Calgary biofilm device model provides fast and reproducible results in determining the minimum biofilm eradication concentration [30]. It is possible to supply dynamic conditions and add fresh nutrients to the formed biofilms, and, after their destruction, it is possible to detect viable cell regrowth.

Each bacteriophage has its spectrum of activity. Therefore, the effect against some bacterial strains may be weak, or bacteria may even be completely resistant to them. One possibility to expand the range of phage activity is to use multi-phage cocktails, as was also done in this study. The second way to improve phage activity is to provide adaptation of phages, which significantly improves their efficiency [15,21]. Pyobacteriophag was initially unable to kill the 01206UR strain, but, after adaptation, a significant improvement in the effect was observed in both the planktonic cells and biofilms (see Figure 5). In complicated cases, when antibiotics do not have the desired effect and the efficiency of phages is low, adaptation can be the solution and cure the patient.

## 4. Materials and Methods

### 4.1. Bacterial Isolates and Antimicrobial Susceptibility

Six *Escherichia coli* strains isolated from urine samples from patients with urinary tract infection (UTI) admitted to Pauls Stradins Clinical University Hospital, Latvia, were used for the study. All strains were identified using VITEK^®^2. Two of the most significant biofilm-formatting strains were selected for further research. Antimicrobial susceptibility for bacterial strains was determined by disk diffusion test and interpreted according to clinical breakpoints of the European Committee on Antimicrobial Susceptibility Testing (EUCAST) v9.0.

### 4.2. Biofilm Screening

Flat-bottom polystyrene microtiter plates with lids (TC Plate 96-well, Suspension, F.Sarstedt, Nümbrecht, Germany) were used in the biofilm-formation assay. The inoculum was prepared using freshly incubated bacterial strains on trypticase soy agar (TSA) plates. Three to five morphologically identical colonies from TSA plates were picked with a sterile loop and transferred to 5 mL of Luria Bertani (LB) broth. The prepared samples were incubated at 37 °C overnight (16 h) and then diluted 1:100. Plates were filled with 200 μL of inoculum in each well, covered with a lid, and incubated at 37 °C for 20 h. Biofilms were stained with crystal violet according to the crystal violet assay protocol, based on the technique described by Stepanovic et al., with minor modifications [31]. After incubation, the liquid from the plates was ejected into the clinical waste container. Using an 8-channel pipette, each well was rinsed twice with sterile 0.9% NaCl solution, and each time the excess solution was shaken out into a clinical waste container. Using a multipipette, 200 μL of 0.1% crystal violet solution was poured into each well and stained for 15 min at room temperature. Subsequently, the plate was washed three times with 200 μL distilled water. Each well was filled with 200 μL of 96% ethanol. With the help of a multipipette, the solution in the wells was mixed to ensure the complete dissolution of the dye. The optical density of each well was measured with an optical spectrophotometer (TECAN INFINITE F50, Männedorf, Switzerland) at a wavelength of 620 nm. The mean OD values of each strain were compared to the cut-off value (ODc), which was defined as three standard deviations above the mean OD of the negative control (LB only). According to these values, all strains were divided into 4 groups: OD ≤ ODc = no biofilm producer; ODc < OD ≤ 2 × ODc = weak biofilm producer; 2 × ODc < OD ≤ 4 × ODc = moderate biofilm producer; 4 × ODc < OD = strong biofilm producer. The results are shown in Table 2.

### 4.3. Bacteriophage Cocktails and Their Propagation

Three commercially available cocktails with phages against *E. coli* were used, Ses and Intesti of ELIAVA Biopreparations Ltd. (Tbilisi, Georgia) and Pyobacteriophag of Microgen Ltd. (Perm, Perm Territory, Russia) [32,33]. The plaque assay was performed using the *E. coli* ATCC 29522 reference strain to determine the phage stock titer. Briefly, phage cocktail tenfold dilutions in 0.9% NaCl were made. Then, 50 μL of diluted phage stock were mixed with 100 μL of freshly grown bacterial suspension in 0.7% molten LB agar, previously incubated in LB broth for 16–18 h at 37 °C, and were poured onto a TSA plate. After overnight incubation at 37 °C, phage plaques on each plate were counted, and the phage titer (PFU/mL) was calculated. To achieve a higher phage cocktail titer, webbed plates of phage titration were used, and 12 mL of LB broth medium was poured on the plate and left at room temperature for 2 h. The supernatant with LB broth and overlay agar was removed and collected in 15 mL plastic tubes. Chloroform (CHCL_3_) was added with the final concentration of 2–3%, and the tubes were mixed and left at 4 °C for 2 h. Subsequently, the tubes were centrifuged at 6000× *g* for 15 min at 4 °C, and the supernatant was filtered using a 0.2 μm filter (Syringe filter, Filtropur S, Sarstedt, Nümbrecht, Germany).

### 4.4. Spot Assay for Phage-Susceptibility Testing

The susceptibility of bacteriophages was detected using the spot test. For the test, the bacteria were grown overnight in LB broth. Approximately 5 mL of melted semi-liquid 0.7% LB semi-liquid melted agar was added to 100 μL of bacterial suspension in a sterile tube. The suspension was gently mixed and poured onto a thin layer of the TSA plate. The plates were left stationary until the upper layer of LB agar was completely solidified. With a pipette, 10 μL of the phage lysate 10^6^ PFU/mL was dropped onto the plates and left in aseptic conditions at room temperature until the drops dried. Subsequently, the plates were incubated overnight at 37 °C. The test results were determined the following day by visually evaluating the effect on the plates and considering it as confluent (+++), semi-confluent (++), overgrown or partial lysis and individual plaques (+), or no lysis (−).

### 4.5. Phage Adaptation Procedure

Three iterative passages of the adaptation procedure, also known as phage training, following modified Appelman’s method [20,34], were executed to increase phage infectious ability and reduce the development of bacterial phage resistance.

To prepare the dilution series of bacteriophage cocktail, 4.5 mL LB broth medium was transferred into 15 mL plastic centrifuge tubes labeled from ‘10^−1^’ until the dilution factor of bacteriophage lysate that was attained after the propagation procedure before. Next, a 0.5 mL aliquot of undiluted bacteriophage suspension was transferred into the first ‘10^−1^’ tube. After the preparation was mixed with the Vortex V-1 plus (Biosan, Riga, Latvia), 0.5 mL was shifted to the next ‘10^−2^’ tube. These steps of tenfold dilution were continued until the last labeled tube was reached. After vortexing, an additional 0.5 mL sample from the last tube was removed.

The infection of the bacterial strain with newly titered phages was ensured by adding 0.05 mL of overnight bacterial culture suspension to each tube containing the titered phage, except the first tube, ‘10^−1^’. This tube served as a negative control tube with phage only. A positive control tube contained exclusively bacterial suspension. After an incubation period at 37 °C for 48 h, the optical density of each tube was measured at a wavelength of 600 nm (OD600). Consequent steps of phage training were applied to the phage tube at the highest dilution, but with OD600 nearer to the negative control. To remove cellular debris, the blend in the relevant tube was treated with 1:50 chloroform (CHCl_3_) and centrifuged (15 min, 6000× *g*, 4 °C). The supernatant obtained was filtered through a 0.2 µm pore size filtration system (Syringe filter, Filtropur S, Sarstedt, Nümbrecht, Germany). The spot test was used to assess the effect of the adapted phage on the corresponding bacterial strain. The phage concentration was determined with the bacterial strain used for the adaptation procedure using a plaque assay.

### 4.6. Minimal Inhibitory Concentration (MIC) and Minimal Biofilm Eradication Concentration (MBEC) Detection

To prepare the inoculum, bacterial strains were incubated overnight (16–18 h) on TSA plates at 37 °C. Bacterial colonies from TSA were suspended in LB broth until an optical density of 1.0 McFarland (3.0 × 10^8^ CFU/mL) was reached and then diluted 1:30 with LB broth (final concertation approximately 1.0 × 10^7^ CFU/mL). Sterile 96-well microplates (Nunc™ MicroWell™ 96-Well, Nunclon Delta-Treated, Flat-Bottom Microplate, Thermo Fisher Scientific, Roskilde, Denmark) were filled with 150 μL of inoculum and sterile LB broth for negative control using a multichannel pipette. Subsequently, plates were covered with a 96-peg lid (Nunc™ Immuno TSP Lids) and secured with Parafilm^®^ (Bemis Company, Inc, Neenah, WI, USA). The plates were incubated at 37 °C for 24 h at 150 rpm. After incubation, the 96-peg lid was transferred to a new 96-well plate filled with 200 μL of serial antibiotic dilution, according to the antimicrobial susceptibility of the strains. Two colons of each plate (1st and 12th) were filled with sterile LB for negative and positive control. Ten columns of the plate were filled with serial dilutions of amoxicillin/clavulanic acid (Amoksiklav, Sandoz, Ljubljana, Slovenia) and ciprofloxacin (Cipralex, KRKA, Novo Mesto, Slovenia).

After 24 h of incubation at 37 °C at 150 rpm, optical density measurements were made at 650 nm to detect minimum inhibitory concentration (MIC) values. The lid with pegs was transferred to the new plate, filled with a sterile 0.9% saline solution, and rinsed for 2 min. To determine the minimal biofilm eradication concentration (MBEC), the peg lid was transferred to a new plate (recovery plate) with 200 μL of sterile LB broth in each well. This plate was sonicated in an ultrasonication bath (Model 08855-02, Cole-Parmer, Vernon Hills, IL, USA) for 30 min at a frequency of 44 Hz to remove the biofilm. After sonification, the lid with pegs was removed, and the plate was covered with a sterile lid and secured with Parafilm^®^. MBEC values were measured at OD650 after 24 h incubation of the recovery plate at 37 °C without rpm.

### 4.7. Colony-Forming Unit Spot Test

The crystal violet assay reflects the formation of the biofilm mass but does not accurately indicate the count of live bacterial cells in the biofilm. The colony-forming unit spot test is used to evaluate the number of viable cells in a broth and to confirm the biofilm-formation process (Figure 3). The colony-forming unit (CFU) spot test was performed to detect biofilm formation after 24 h and 48 h. The peg with biofilm was cut from the lid with sterile pliers, transferred to 200 μL of sterile saline, and sonicated for 30 min. Subsequently, the bacterial suspension was serially diluted, and 10 μL of each concentration was placed on TSA plates. After 16–20 h of incubation at 37 °C, the colonies were counted, and the CFU values were calculated.

### 4.8. Data Analysis

All experiments were performed in duplicate. All data were expressed as mean values. Microsoft Excel 10 and GraphPad Prisma 9.4.1 were used for data analysis. Mann–Whitney U test was used to compare the differences between control and treatment groups.

### 4.9. Ethical Statement

The study protocol was approved by the ethics committee of Riga Stradins University (document no. 8/08.09.2016).

## 5. Conclusions

Our results indicate that bacteriophages can successfully destroy bacterial biofilms, even when antibiotics cannot. Despite this, bacteriophages cannot be used in all situations, so each case must be evaluated individually because we lack controlled clinical studies that determine the effect of phages in biofilm-associated infections. Bacteriophages are promising agents for biofilm-associated infections, so further studies are needed to investigate the action of bacteriophages in bacterial biofilms more closely. Future research should also focus on the phage–antibiotic interaction in planktonic and biofilm-forming cells.

## Figures and Tables

**Figure 1 antibiotics-11-01706-f001:**
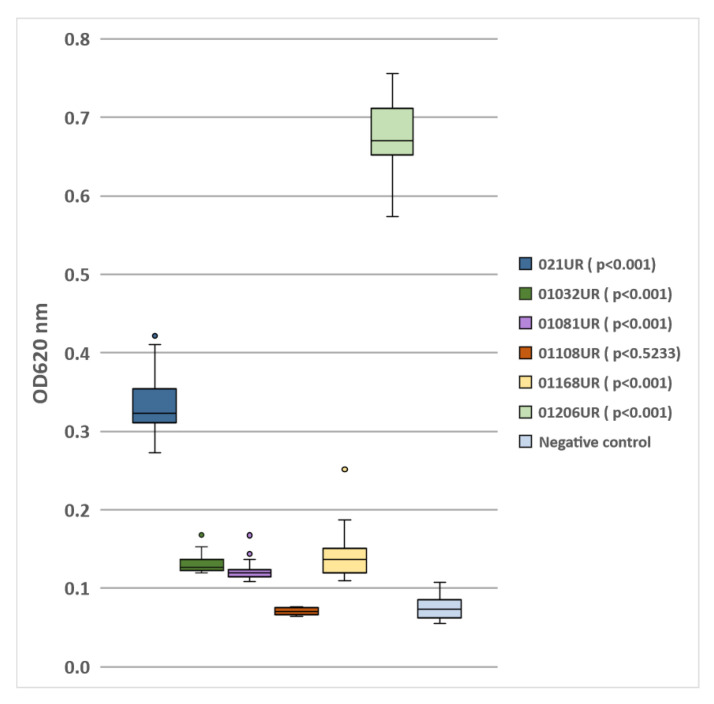
Biofilm formation by *E.coli* isolated from patients’ urine samples.

**Figure 2 antibiotics-11-01706-f002:**
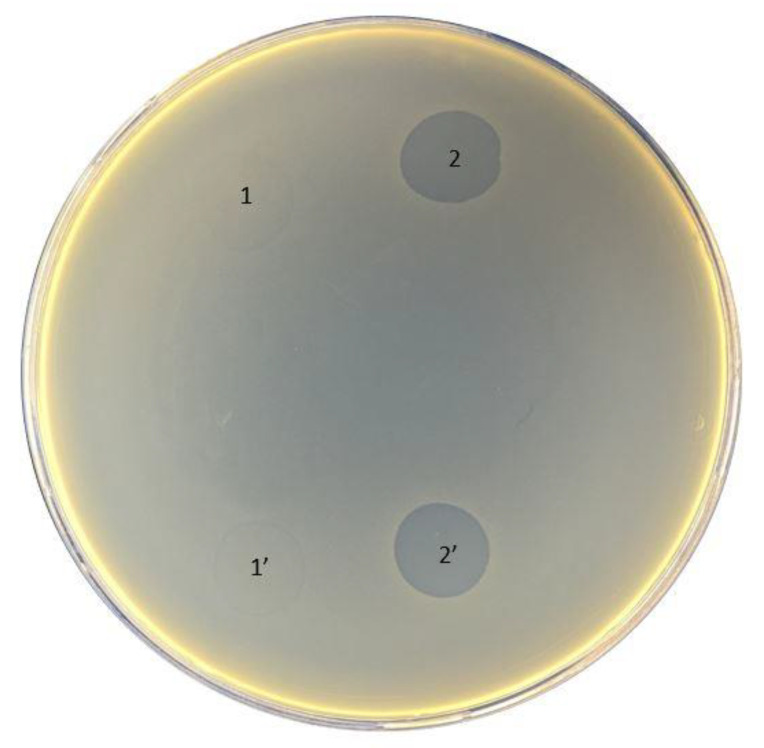
Pyobacteriophag effect before (1 and 1′) and after (2 and 2′) adaptation to strain 01206UR.

**Figure 3 antibiotics-11-01706-f003:**
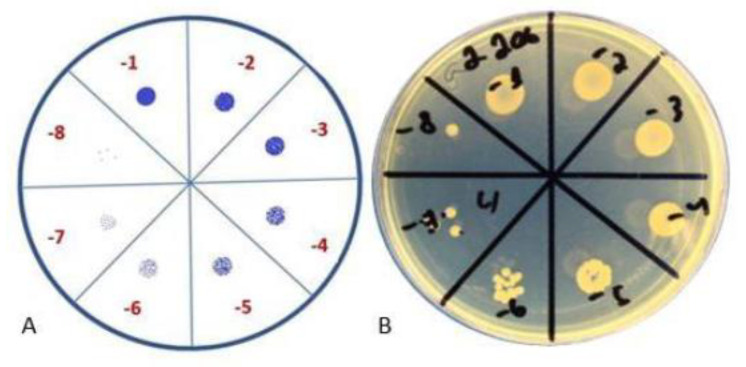
Colony-forming unit spot test. (**A**) Schematic representation; (**B**) visual representation.

**Figure 4 antibiotics-11-01706-f004:**
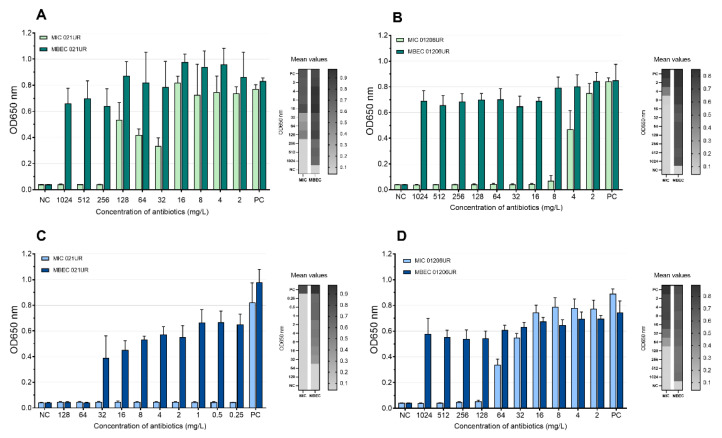
Mean MIC and MBEC antibiotic values. (**A**) Amoxicillin–clavulanic acid for 021UR; (**B**) amoxicillin–clavulanic acid for 01206UR; (**C**) ciprofloxacin for 021UR; (**D**) ciprofloxacin for 01206UR; NC—negative control; PC—positive control.

**Figure 5 antibiotics-11-01706-f005:**
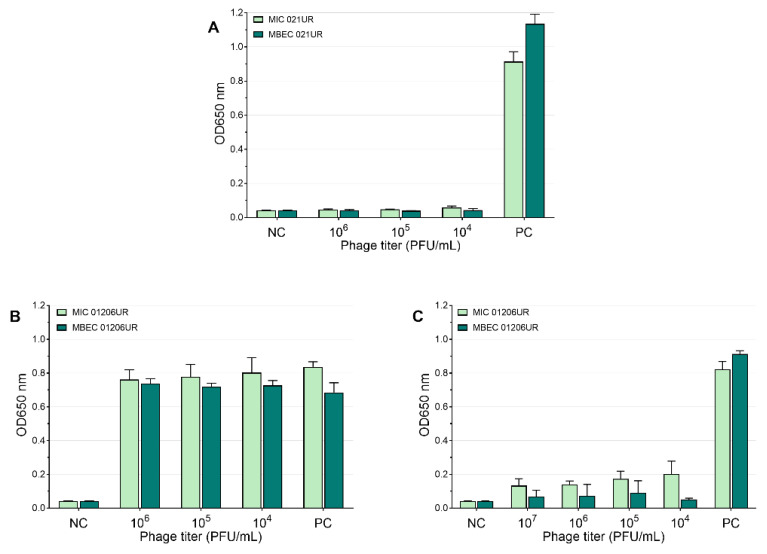
Pyobacteriophag effect at different concentrations represented as mean values. (**A**) MIC and MBEC in 021UR; (**B**) MIC and MBEC in 01206UR before adaptation; (**C**) MIC and MBEC in 01206UR after adaptation.

**Figure 6 antibiotics-11-01706-f006:**
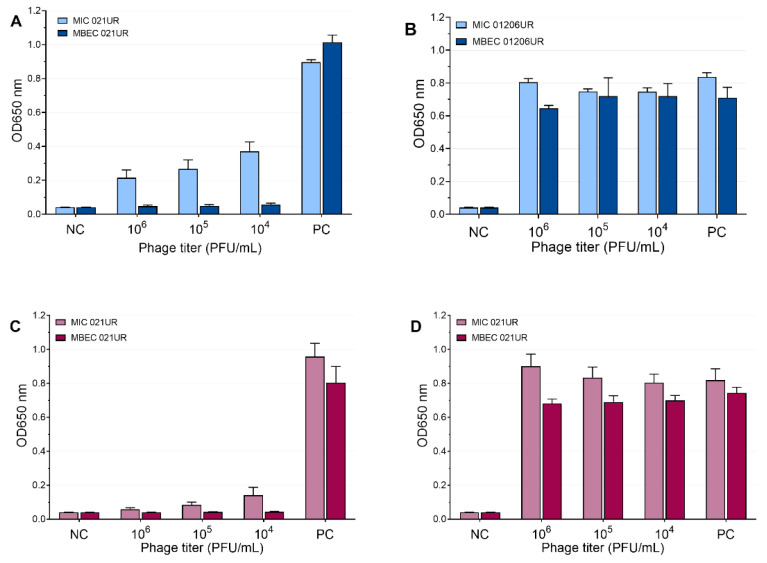
Ses and Intesti phage effects at different concentrations represented as mean values. (**A**) MIC and MBEC in 021UR by SES; (**B**) MIC and MBEC in 01206UR by Ses; (**C**) MIC and MBEC in 021UR by Intesti; (**D**) MIC and MBEC in 01206UR by Intesti.

**Table 1 antibiotics-11-01706-t001:** Patient characteristics for isolated bacterial strains. CKD, chronic kidney disease; FSGS, focal segmental glomerulosclerosis; CMV, cytomegalovirus; UTI, urinary tract infection.

N	Strain Code	Age	Sex	Type of UTI	Comorbidities
1	021UR	21	Female	Pyelonephritis	Non
2	01032UR	66	Male	Pyelonephritis	Diabetes mellitus; CKD IIIB; Nephrolithiasis; Essential arterial hypertension; Coronary artery disease; Congestive heart failure
3	0108UR	30	Female	Asymptomatic bacteriuria	Renal transplantation 2013; CKD V; FSGS; Haemodialysis; Renal anaemia; Secondary hypertension
4	01108UR	74	Female	Cystitis	Coronary artery disease; Essential arterial hypertension; Osteoarthritis; Hypothyroidism; Congestive heart failure
5	01168UR	49	Female	Cystitis	Renal transplantation 2016; Acute CMV hepatitis; Secondary hypertension; Multiple sclerosis;
6	01206UR	57	Female	Recurrent cystitis	Renal transplantation 2008; CKD IIIB; Recurrent UTI; Secondary hypertension;

**Table 2 antibiotics-11-01706-t002:** Results of antimicrobial and phage-susceptibility testing and ability of biofilm formation.

	021UR	01032UR	01081UR	01108UR	01168UR	01206UR
AMP	R	R	S	S	R	R
AMC	R	R	S	S	S	S
TZP	S	I	S	S	S	S
CTX	S	R	S	S	S	S
CAZ	S	R	S	S	S	S
IMP	S	S	S	S	S	S
MEM	S	S	S	S	S	S
CIP	S	R	S	S	S	R
NOR	S	N/A	N/A	N/A	S	R
GEN	S	R	S	S	S	S
SXT	R	S	S	S	R	S
NIT	S	S	S	S	S	S
ESBL	Negative	Positive	Negative	Negative	Negative	Negative
Ses	++	+	++	+	++	-
Intesti	+++	+	++	-	+	-
Pyobacteriophag	++	+	+	-	+	+
Pyobacteriophag *	N/A	N/A	N/A	N/A	N/A	+++
Biofilm formation	Moderate	Weak	Weak	No biofilm	Weak	Strong

S—sensitive; R—resistant, I—intermediate; ampicillin (AMP); amoxicillin–clavulanic acid (AMC); piperacillin–tazobactam (TZP); cefotaxime (CTX); ceftazidime (CAZ); imipenem (IMP); meropenem (MEM); ciprofloxacin (CIP); norfloxacin (NOR); gentamicin (GEN); sulfamethoxazole–trimethoprim (SXT); nitrofurantoin (NIT); extended-spectrum beta-lactamase (ESBL); * Pyobacteriophag phage cocktail after adaptation; not applicable (N/A).

**Table 3 antibiotics-11-01706-t003:** Results of colony-forming unit spot test.

Bacterial Strain	Inoculum	Planktonic Cells (after 24 h)	Biofilm Formatting Cells (after 24 h)	Biofilm Formatting Cells (after 48 h)
021UR	10^7^ CFU/mL	10^9^ CFU/mL	10^6^ CFU/mL	10^7^ CFU/mL
01206UR	10^7^ CFU/mL	10^9^ CFU/mL	10^6^ CFU/mL	10^7^ CFU/mL

## Data Availability

The data presented in this study are available on request from the corresponding author.

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
