# Peer review of "Anti-Biofilm Effect of Bacteriophages and Antibiotics against Uropathogenic Escherichia coli"

_antibiotics, 2022, doi:10.3390/antibiotics11121706_

Round 1

Reviewer 1 Report

The manuscript on "Anti-biofilm effect of bacteriophages and antibiotics against uropathogenic Escherichia coli" addresses a very pertinent problem of treating infections caused by antibiotic resistant bacteria. Using bacteriophage cocktails for bacteria that are resistant to antibiotics offers an effective solution for treatment of MDR infections. However, the research requires further detailing such as optimization of bacteriophage control process. Nevertheless, in this manuscript authors have presented a well-designed research with the use of appropriate methodologies and well-presented results.

The manuscript is highly recommended for publication.

Author Response

Thank you very much for time and the effort you have spent reviewing the manuscript. Our research group is thankful for your recommendation.

Reviewer 2 Report

The work presented is well written and relevant to the topic it proposes to address. Only a few adjustments are needed in the presentation of the result tables.

Author Response

Thank you very much for time and the effort you have spent reviewing the manuscript!

Reviewer 3 Report

Article Summary

            With antimicrobial resistance on the rise and increased failure rates of current antimicrobials, it is essential to find alternative therapies to combat AMR bacteria. The authors’ work represents an area of research that is of critical importance. The authors received six Escherichia coli (E. coli) samples from patients with urinary tract infections (UTIs). These six strains were tested for biofilm production and for sensitivity against different antibiotics and three different bacteriophage cocktails (Pyobacteriophage, Ses, Intesti) that have already been published. Next, the reduction in biofilm was examined after treatment with antibiotics or phage cocktail. The authors conclude that bacteriophages can reduce biofilm production caused by microorganisms.

Aim: Test bacteriophage and antibiotics against biofilm-producing E. coli isolates

Major Contributions: Additional biofilm activity information concerning the three phage cocktails used

General Questions Summary

Is the manuscript clear, relevant for the field and presented in a well-structured manner?

             The manuscript itself is easy enough to follow. There are slight grammatical and formatting errors, but these do not make it impossible to follow the flow of the paper. My concern is the conclusion of the paper: “Our results indicate that bacteriophages can successfully destroy bacterial biofilms…” I think that the experiments in this paper should be expanded upon (include more strains, perform CFU/mL assays after treatment, etc.) to be more relevant for the field. Antibiofilm activity by bacteriophages is already known. Rephrasing for the importance towards these particular strains may be beneficial.

Are the cited references mostly recent publications (within the last 5 years) and relevant? Does it include an excessive number of self-citations?

            Yes, the majority of citations are recent publications. Excessive self-citation does not occur.

Is the manuscript scientifically sound and is the experimental design appropriate to test the hypothesis?

            The manuscript itself is scientifically sound, but it can be improved upon with the addition of additional experiments to confirm findings.

Are the manuscript’s results reproducible based on the details given in the methods section?

            The results should be somewhat reproducible based on the details given in the methods section. There are some slight inconsistencies between the methods and the results listed in the paper (such as what optical density the cells were measured at). If this paper is to be published, these mistakes need to be rectified.

Are the figures/tables/images/schemes appropriate? Do they properly show the data? Are they easy to interpret and understand? Is the data interpreted appropriately and consistently throughout the manuscript? Please include details regarding the statistical analysis or data acquired from specific databases.

             Figures are appropriate for their respective experiments. Labeling of the figures appears to contradict the method section, however. No statistical analyses provided for figures.

Are the conclusions consistent with the evidence and arguments presented?

            Conclusions are generic and broad, relaying known information. Sample size is too small.

Please evaluate the ethics statements and data availability statements to ensure they are adequate.

            No issues with ethical statements.

Major Revisions

1.    For Figure 1, how many times was biofilm screening performed? With the exception of 01108UR, the other bacterial strains do appear to at least be somewhat higher. Are these results significant? I understand that the two strains with the highest optical density reading were selected, but why were only two strains selected and not the others to expand the treatment groups? I do not think two strains would be sufficient to determine a general applicability of bacteriophage to biofilm, especially when the differences between each strain is highlighted in this figure. Remove comma in Y axis label.

2.    For Table 2, why was bacteriophage adaptation only applied to 1206UR and not to another strain like 01108UR? According to the results listed, both are resistant to two out of three cocktails and are only partially sensitive to the third. Was any experiment performed (besides the MIC/MBEC experiment later) to show phage adaptation? For instance, was the cocktail that underwent adaptation tested against other strains as well? Was it an adaptation to that host or simply an increase in titer? Were any images taken for the phage susceptibility screen so the differences between the rankings can be observed?

3.    For Table 3 and Figure 2, was the colony forming unit assay performed for any experiments later in the manuscript? I think it would be wise to show these data in addition to the optical density readings. The liquid media used for this particular experiment is not listed in the methods section. Positive control is improperly described in the methods section. The optical density at which the experiment is read is listed as 650 nm in the methods, but the figure lists 600 nm on the Y axis. Which was used?

4.    For Figures 3, 4, 5, why was the mean MIC/MBEC not calculated for all figures? Again, there is a discrepancy between the Y axis and the methods section for the optical density measurement used. How many times were these experiments performed? Is there any significance with the findings?

5.    Methods section needs to be reformatted to have better separation between paragraphs and sections. Subsection labels should stand out in some way to make reading/finding one’s place much easier.

6.    In reference to the Discussion and manuscript overall, if the authors had access to antibiotics and bacteriophage cocktail, why was an experiment with the two in combination not conducted? After showing supposed prevention of biofilm formation with bacteriophage, using the two in combination could potentially yield better results, particularly in CFU/mL assays.

Minor Revisions

1.    I was only able to find a published cocktail under the name Pyobacteriophage. If this is the same cocktail, it appears to be misspelled throughout the manuscript.

2.    Grammatical and formatting issues occur throughout the paper. It does not keep the paper from being understood, but it does hinder reading somewhat.

Round 2

Reviewer 3 Report

Thank you for the responses to my comments. Though I would still like to see additional CFU data, I can understand the complications. Overall, this version of the manuscript is much better than the original. There are some slight changes that need to be made, like moving Results in line 91 to the next page (it looks a little odd to me just floating there), but other than that, I would recommend this for publication.

Author Response

Thank you very much for your time and the effort you spent improving the manuscript! Thank you for the note! Formatting will be corrected in the final version.